# Acute Inflammation Is a Predisposing Factor for Weight Gain and Insulin Resistance

**DOI:** 10.3390/pharmaceutics14030623

**Published:** 2022-03-11

**Authors:** Edson Mendes de Oliveira, Jacqueline C. Silva, Thais P. Ascar, Silvana Sandri, Alexandre F. Marchi, Silene Migliorini, Helder T. I. Nakaya, Ricardo A. Fock, Ana Campa

**Affiliations:** 1Departamento de Análises Clínicas e Toxicológicas, Faculdade de Ciências Farmacêuticas, Universidade de São Paulo, Av. Prof. Lineu Prestes, 580, São Paulo 05508-000, Brazil; jcaval4@uic.edu (J.C.S.); thaispalumbo@hotmail.com (T.P.A.); ssandris@gmail.com (S.S.); alexandrefmarchi@gmail.com (A.F.M.); sika@usp.br (S.M.); hnakaya@usp.br (H.T.I.N.); hemato@usp.br (R.A.F.); 2Hospital Israelita Albert Einstein, Av. Albert Einstein, 627, São Paulo 05652-900, Brazil

**Keywords:** infection, adipocyte, SAA, TLR-4, TLR-2, CD14

## Abstract

In the course of infection and intense endotoxemia processes, induction of a catabolic state leading to weight loss is observed in mice and humans. However, the late effects of acute inflammation on energy homeostasis, regulation of body weight and glucose metabolism are yet to be elucidated. Here, we addressed whether serial intense endotoxemia, characterized by an acute phase response and weight loss, could be an aggravating or predisposing factor to weight gain and associated metabolic complications. Male Swiss Webster mice were submitted to 8 consecutive doses of lipopolysaccharide (10 mg/kg LPS), followed by 10 weeks on a high-fat diet (HFD). LPS-treated mice did not show changes in weight when fed standard chow. However, when challenged by a high-fat diet, LPS-treated mice showed greater weight gain, with larger fat depot areas, increased serum leptin and insulin levels and impaired insulin sensitivity when compared to mice on HFD only. Acute endotoxemia caused a long-lasting increase in mRNA expression of inflammatory markers such as TLR-4, CD14 and serum amyloid A (SAA) in the adipose tissue, which may represent the key factors connecting inflammation to increased susceptibility to weight gain and impaired glucose homeostasis. In an independent experimental model, and using publicly available microarray data from adipose tissue from mice infected with Gram-negative bacteria, we performed gene set enrichment analysis and confirmed upregulation of a set of genes responsible for cell proliferation and inflammation, including TLR-4 and SAA. Together, we showed that conditions leading to intense and recurring endotoxemia, such as common childhood bacterial infections, may resound for a long time and aggravate the effects of a western diet. If confirmed in humans, infections should be considered an additional factor contributing to obesity and type 2 diabetes epidemics.

## 1. Introduction

Metabolic endotoxemia is low-grade endotoxemia derived from intestinal microbiota and modulated by high-fat feeding. This condition is known to contribute to weight gain and insulin resistance. [1,2]. On the other hand, transient and intense endotoxemia is observed along bacterial infections, and it is characterized by a high catabolic process that frequently leads to weight loss [3]. However, late effects of acute inflammation on the adipose tissue remain to be fully explored.

Experimental endotoxemia can be achieved by intravenous or intraperitoneal administration of high doses of lipopolysaccharide (LPS), triggering a pronounced acute inflammatory response similar to that observed during Gram-negative bacteria infections [4]. Acute inflammatory response is regulated by a variety of endogenous factors [5], starting with LPS binding to several host soluble and cell-surface molecules, such as cluster of differentiation 14 (CD14), expressed on the plasma membrane of monocytes and macrophages [6]. CD14 is known to chaperone LPS molecules to the toll-like receptor 4 (TLR4) signaling complex, triggering a cascade of events regulating the production of acute phase proteins by the liver and a multitude of inflammatory mediators in nearly all tissues and organs. Serum amyloid A (SAA) and C-reactive protein (CRP) are examples of increased inflammatory proteins in the circulation and are commonly used in clinical practice as unspecific markers of an acute phase response [4,7,8]. The hallmarks of intense endotoxemia also include endothelial activation, changes in subsets of circulating leukocytes and peripheral macrophage activation [5,9].

Although acute inflammation is promptly associated with weight loss, we hypothesized that transient and intense endotoxemia promotes permanent changes in the adipose tissue, which might predispose adipose hypertrophy and ultimately weight gain, especially when challenged with an obesogenic stimulus. In order to assess that, we combined an experimental acute endotoxemia model (consecutive LPS challenges), followed by an established diet-induced obesity protocol in mice. Additionally, for the purpose of verifying if acute infections are associated with changes in the adipose tissue, including cell proliferation, adipogenesis or inflammation, we performed gene set enrichment analysis using a publicly available database relative to the effect of Gram-negative bacteria infections on adipose tissue.

If intense endotoxemia leads to biochemical changes involved in adipose tissue hyperplasia and hypertrophy, acute inflammation might be considered an aggravating factor for weight gain and obesity comorbidities. Collectively, this could be of additional relevance for developing countries, where recurrent bacterial infections are more prevalent and obesity rates are rising fast [10,11,12].

## 2. Materials and Methods

### 2.1. Animals

Male Swiss Webster mice (21 days of age) were obtained from the Animal Facility of the Faculty of Pharmaceutical Sciences, University of São Paulo, Brazil, under approval by its Ethical Committee (CEEA No. 297, 7 February 2011). The animals were housed inside standard polypropylene cages in a room maintained at 22 ± 2 °C in a 12:12 h light/dark cycle (lights on at 7:00 a.m. and off at 7:00 p.m.) and relative humidity of 55 ± 10%. Body weight was measured once a week during the entire protocol. Food and water intake were kept ad libitum and were measured every 2 days. Euthanasia occurred by anesthesia overdose and was ensured by cervical dislocation.

### 2.2. Acute Endotoxemia Followed by Recovery Period under Chow Diet

The method of multiple inductions of acute endotoxemia comprises intraperitoneal administration of 8 consecutive injections (every 3 days) of 10 mg/kg LPS (lipopolysaccharides from *Escherichia coli* 026:B6, Sigma-Aldrich^®^, St. Louis, MO, USA), in saline (0.9% NaCl; Sigma-Aldrich^®^, St. Louis, MO, USA), starting at weaning (21 days of age) with the end at 45 days of age of the animal. The time between two acute endotoxemia (3 days) was defined by the profile of the SAA serum levels. It was observed that after LPS injection, SAA concentration increased over 100 times, peaking at 12 hours (approximately 1500 µg/mL) and returning to basal levels at 72 hours (Appendix A, Appendix A). For acute endotoxemia experiments, mice were randomly assigned into 2 different groups, control and LPS, with euthanasia occurring after the last acute phase period or after 6 weeks from the last acute phase induction (recovery period). The experimental design is illustrated in Appendix A (Appendix A).

### 2.3. Acute Endotoxemia Followed by High-Fat Diet (HFD)

For experiments of acute endotoxemia followed by 10 weeks on a high-fat diet (LPS + HFD), the animals were randomly assigned into 2 different groups, HFD and LPS + HFD. The HFD mice were submitted to a HFD for 10 weeks, starting concurrently with the LPS + HFD group. The LPS + HFD mice underwent multiple inductions of acute endotoxemia, followed by 1 week of recovery period in a standard chow diet plus 10 weeks on a 30% HFD. In our experimental model, we consider the recovery period 7 days, defined according to weight reestablishment. The diet was produced following the American Institute of Nutrition’s recommendations for the adult rodent, and its composition is listed in Appendix A. Body weight was measured every 3 days during the acute phase period. The experimental design is illustrated in Appendix A.

### 2.4. Glucose and Insulin Tolerance Tests and Measurements of Serum Leptin, Adiponectin, Insulin, IGF-I, SAA and Endotoxin

Intraperitoneal glucose and insulin tolerance tests (IPGTT and IPITT) were performed as described previously [13]. Serum concentrations of the proteins below were determined using ELISA following the manufacturer's instructions: leptin, adiponectin and insulin (Millipore^®^ Corporation, Billerica, MA, USA), SAA (Tridelta Development Ltd., Maynooth, Ireland) and IGF-I (R&D Systems^®^, Minneapolis, MN, USA). Endotoxin was measured with the Limulus Amoebocyte Lysate (LAL) chromogenic end-point assay (Lonza^®^, Allendale, NJ, USA).

### 2.5. Histological Analysis

Paraffin-embedded sections (5 μm thick) from epididymal adipose tissue were stained with hematoxylin and eosin to assess morphology. Immunohistochemistry for F4/80^+^ was performed using a rat anti-mouse F4/80^+^ antibody (1:100 dilution, AbD Serotec^®^, Raleigh, NC, USA) subsequently incubated with the appropriate secondary biotinylated antibody (Vector Laboratories Inc., Burlingame, CA, USA) and visualized with ImmPACT AEC peroxidase (Vector Laboratories Inc., Burlingame, CA, USA). Immunofluorescence for F4/80^+^, SAA and perilipin was performed using a rat anti-mouse F4/80^+^ antibody and rabbit anti-mouse perilipin (both 1:100 dilution, Abcam^®^, Cambridge, UK) and a rabbit anti-mouse SAA antibody (1:200 dilution, kindly provided by Dr. de Beer Laboratory, University of Kentucky, KY, USA), subsequently incubated with the appropriate secondary fluorescent antibody (Invitrogen^®^, Camarillo, CA, USA). Slides were mounted using Vectashield set mounting medium with 4,6-diamidino-2-phenylindole-2-HCl (DAPI) (Vector Laboratories Inc., Burlingame, CA, USA). An isotype control was used to ensure antibody specificity in each staining. Tissue sections were observed with a Nikon Eclipse 80i microscope (Nikon^®^, Tokyo, Japan), and digital images were captured with NIS-Element AR software (Nikon^®^).

### 2.6. In Vivo Peripheral Fat Area Quantification

Two X-ray images were taken at different energy levels using the MS FX Pro multispectral imaging system (Carestream Health Inc., Woodbridge, CT, USA), allowing the adipose tissue to be mapped out on the animals as described previously [13].

### 2.7. Quantitative Real-Time PCR

Total RNA from epididymal adipose tissue was isolated using the Qiagen RNeasy^®^ Lipid Tissue Mini kit (Qiagen, Hilden, Germany). cDNA was then synthesized from 1 µg of RNA using the High Capacity cDNA Reverse Transcription kit (Life Technologies^®^, Grand Island, NY, USA). Real-time PCR was performed using SyBr^®^ Green Master Mix (Life Technologies^®^, Grand Island, NY, USA). The primer sequences are detailed in Appendix A (Appendix A). Real-time PCR for *Saa3* was performed using the TaqMan^®^ assay (Applied Biosystems^®^, Grand Island, NJ, USA), Catalog Number Mm00441203_m1, *Saa3 a*nd β-actin (*Actb*), Number 4552933E, as an endogenous housekeeping gene control. Relative gene expression was determined using the 2^–∆∆Ct^ method.

### 2.8. Gene Set Enrichment Analysis of Publicly Available Microarray Data

We collected from GEO (http://www.ncbi.nlm.nih.gov/geo, GSE50647, date of access on 13th October 2014) the expression profiles of mouse visceral adipose tissue. In Reference [14], the authors exposed chow-fed apolipoprotein E (apoE)-deficient mice to either (1) recurrent intravenous infection with *A. actinomycetemcomitans* or (2) a combination of recurrent intravenous infection with *A. actinomycetemcomitans* with chronic intranasal infection with *C. pneumonia*. For the gene set enrichment analysis (GSEA), we ranked genes based on their mean log_2_ fold-change values between infected versus uninfected mice. We then utilized custom gene sets, which contained genes related to proliferation, adipogenesis, inflammation and SAA family. GSEA was performed using default parameters. Heat maps were used to display all genes from a statistically significant gene set.

### 2.9. Statistical Analysis

Results were presented as mean ± SEM. Statistical analysis was performed with GraphPad Prism4 (Graph Pad Software, Inc., San Diego, CA, USA). Comparisons between two groups were conducted with the unpaired Student’s *t* test. Data with two independent variables were tested by two-way analysis of variance with Bonferroni post hoc test. The level of significance was set at **p* < 0.05.

## 3. Results

### 3.1. Acute Endotoxemia Affects Adipose Tissue but Does Not Lead to Weight Gain in Mice under Chow Diet

In order to verify the effects of acute endotoxemia on adipose tissue, mice were subjected to eight consecutive LPS challenges (Appendix A, Appendix A). During the acute phase response, serum endotoxin (Figure 1a) and SAA (Figure 1b) levels raised over a hundredfold, and mice developed overt signs of endotoxemia (hunched posture, reduced movement and piloerection), with no animal death. It is known that acute endotoxemia leads to reduced food intake and weight loss. During the acute endotoxemia period, LPS-treated animals showed a reduced caloric intake (up to 40%; Figure 1c), associated with weight loss (approximately 12.5% of their total weight; Figure 1d) and 20% reduction of epididymal adipose tissue mass (Figure 1e).

Histological analysis from the epididymal adipose tissue (Figure 1l) showed a 30% decrease in adipocyte size in the LPS group (Figure 1f). Besides weight loss, the adipose tissue presented increased inflammatory markers, such as macrophage infiltration (F4/80^+^ cells, Figure 1l) and *Saa3* mRNA expression (Figure 1h), without *Saa1.1* and *Saa2.1* modulation in the adipose tissue (Figure 1g). The expression of *Tlr-4* (Figure 1j) and *Cd14* (Figure 1k) had an increase of more than three times, although the *Tlr-2* (Figure 1i) remained unaltered.

No tolerance to LPS was observed during the experimental protocol, with similar increment profiles in serum endotoxin (Appendix A, Appendix A) and SAA (Appendix A, Appendix A) after each LPS challenge. After the endotoxemia period, the animals recovered their weight in the course of one week and presented comparable growth curves when compared to the control group in the 42 consecutive days (6 weeks) (Figure 1c,d). Serum levels of endotoxin (Figure 1a) and SAA (Figure 1b) were back to basal levels after a week with the animals in recovery (LPS + REC). However, the expression mRNA levels of *Saa3* (Figure 1h), *Tlr-4* (Figure 1j) and *Cd14* (Figure 1k) remained increased in the adipose tissue.

### 3.2. A Previous History of Acute Endotoxemia Exacerbates High-Fat Diet Complications

After the end of LPS challenges followed by a one-week recovery period, mice were submitted to a HFD for 70 days (10 weeks) (Figure 1b, Appendix A). HFD-fed mice without previous LPS treatment were used as controls. The shift of chow diet to HFD resulted in an increment of approximately twice in the caloric intake for both groups, HFD and LPS + HFD (Figure 2a). Despite the similar caloric intake between these groups, mice previously submitted to multiple LPS challenges (LPS + HFD) showed a distinct growth curve with a 15% increase in total body weight (Figure 2b), with increased epididymal (Figure 2c) and subcutaneous (Figure 3e) adipose tissue depots and no difference in retroperitoneal (Figure 3d) fat. The data were confirmed using X-rays images highlighting the subcutaneous fat area on the animals, showing that LPS + HDF mice have a larger peripheral fat tissue, with an increment of 23% in the adipose tissue area (Figure 2f,g). Besides that, epididymal fat from LPS + HFD mice presented larger adipocytes when compared to the HFD group (Figure 2h), which may explain differences in their metabolic phenotype. On the other hand, there is no difference in the adipose tissue inflammatory profile, showing similar macrophage infiltration (F4/80^+^) and SAA production (Figure 2i) in both HFD and LPS + HFD groups. However, it is important to highlight that all aforementioned parameters (adipocyte hypertrophy, macrophage infiltration and SAA production in the adipose tissue) are increased when compared to the lean control group (under chow diet) (Figure 2i).

### 3.3. Acute Endotoxemia, Inflammatory Markers and Glucose Homeostasis

As an effect of the high-fat diet, a 2-fold increase in serum endotoxin and SAA levels was observed in the HFD and LPS + HFD groups when compared to lean mice (Figure 3a,b compared to Figure 1a,b; respectively). However, no differences were observed between HFD and LPS + HFD groups. In a similar manner, transcript levels of *Saa3* in the adipose tissue were increased in response to the HFD, regardless of previous treatment with LPS (Figure 3d compared to Figure 1h). mRNA expression levels of *Saa1.1*/*Saa2.1* were similar among all conditions (Figure 3c compared to Figure 1g). Nevertheless, mice previously submitted to multiple LPS challenges showed an increment in *Tlr-4* (Figure 3f) and *Cd14* (Figure 3g) of mRNA expression in the adipose tissue, with no change in *Tlr-2* transcript levels (Figure 3e). Additionally, mice previously submitted to multiple acute endotoxemia had a worsened metabolic profile after the high-fat diet period, showing increased leptin (Figure 3h) and insulin (Figure 3l) levels, with impaired glucose homeostasis, as measured by glucose and insulin tolerance tests (Figure 3m,n*)*. Serum adiponectin (Figure 3i), IGF-1 (Figure 3j) and fasting glucose (Figure 3k) concentrations were also measured, and no significant differences were observed.

### 3.4. Recurrent Infection Upregulates Proliferative and Inflammatory Genes in Adipose Tissue

We looked at the GEO database for studies in mice where transcriptome analysis was performed in mice adipose tissue after any sort of acute inflammatory process. From the study GSE50647 [14], where mice were infected with Gram-negative bacteria (*A. actinomycetemcomitans* or coinfected with *A. actinomycetemcomitans* and *C. pneumonia)*, it was observed that a group of genes responsible for driving proliferation and inflammation was upregulated after acute infection, as well as SAA-related genes (SAA isoforms and receptors). On the other hand, the cluster of genes involved in adipogenesis was downregulated (Figure 4).

## 4. Discussion

Obesity and associated comorbidities, such as type 2 diabetes, represent a major global public health concern. A novel and more detailed mechanistic understanding of the molecular, physiological and behavioral pathways involved in the development of obesity are critical for tackling this epidemic [15]. Here, we report that multiple and intense endotoxemia causes long-lasting biochemical alterations in the adipose tissue that may predispose and intensify the harmful effects of a high-fat diet.

The experimental model of intense and transient endotoxemia used in this study led to an approximately 150-times increment in serum endotoxin levels, reaching values near to 300 EU/mL, which is comparable to levels found in mice and humans during infectious diseases [16]. Endotoxemia induces the release of a large amount of inflammatory mediators, such as proinflammatory cytokines and highly reactive oxygen and nitrogen intermediates [17,18]. Serum amyloid A, a cytokine-like protein involved in the propagation of the acute phase response [19], is increased nearly 1000 times after each dose of LPS before returning to its basal levels in 72 hours (Appendix A, Appendix A). During the endotoxemic phase, mice showed a decrease in food intake and a marked depletion in fat depots. After the termination of LPS challenges, the animals showed a rapid weight recovery and a similar growth curve pattern when compared to the control group (Figure 1). However, other inflammatory markers significantly increased in the adipose tissue after the series of LPS challenges, including macrophage infiltration, *Saa3*, *Tlr-4* and *Cd14* mRNA expression, remaining persistently elevated even after 6 weeks of recovery. The activation of TLR-4 and its coreceptor CD14 are associated with obesity and insulin resistance in mice [1,20,21,22,23] and may represent one of the key elements connecting inflammation to increased susceptibility to weight gain and impaired glucose homeostasis. In fact, mice treated with LPS followed by a high-fat diet showed an approximately 15% increase in weight and impaired insulin sensitivity when compared to mice only subjected to the high-fat diet (Figure 2 and Figure 3). Increased serum leptin levels, as a proxy of the degree of adiposity, were also observed in the LPS-HFD mice. Leptin regulates adipose tissue mass through central hypothalamus mediated effects on food intake, and increased serum leptin levels in obesity are often associated with a failure of the feedback loop and central leptin resistance [15].

TLR-2 is another member of the toll-like receptors family with an established role in immune response and is also associated with metabolism and energy homeostasis. In mice, it is known that TLR-2 deficiency confers protection against insulin resistance and also reduces HFD-induced tissue inflammation, in a process dependent on the host microbiota [24]. Further, although no differences were observed in the expression levels of TLR-2 in our experimental model of acute endotoxemia, it is conceivable to speculate that the interaction between TLR-2 and TLR-4 might be affected, modulating its receptor cooperation and further cellular response to LPS [25].

The mechanisms by which acute endotoxemia may lead to a more severe phenotype arising from a HFD are undoubtedly complex and may encompass many factors, such as endogenous TLR-4 and TLR-2 ligands. For instance, SAA is a TLR-2 and TLR-4 agonist [26,27], with serum levels positively correlated with body fat in humans [28]. Whilst the liver is the major site for SAA synthesis, the adipose tissue also contributes to circulating levels of SAA, in a process modulated under hypoxic conditions [29,30], a cardinal event during adipose tissue expansion. SAA production is associated with a proinflammatory response and cellular hyperplasia [27,31,32,33,34,35,36,37,38,39,40]. In addition, data from SAA KO mice [40,41] and studies using SAA antisense oligonucleotide (ASO_SAA_) [42,43] strongly suggest that SAA is part of the LPS signaling pathway, associating inflammation with obesity and insulin resistance.

For the purpose of evaluating the comprehensiveness of some of our findings, and to address our hypothesis that acute inflammation may induce preadipocyte proliferation while triggering SAA production, we performed gene set enrichment analysis in publicly available microarray data (GSE50647) [14]. We identified that whilst different clusters of genes responsible for driving cell proliferation, inflammation and the expression of SAA-related proteins (SAA isoforms and receptors) were upregulated, genes involved in adipogenesis were found to be downregulated in the adipose tissue of mice previously infected with Gram-negative bacteria. By increasing preadipocyte proliferation within the adipose tissue and possibly aggravating its susceptibility to an obesogenic diet, the process of infection might provide the appropriate condition for an exacerbated outcome of weight gain and insulin resistance.

Epidemiological data show that low-income countries experience a higher prevalence of infectious diseases in children, including acute diarrheal disease, neonatal sepsis and severe bacterial infections [44,45,46]. In a similar context, social inequality and poverty are strongly associated with childhood obesity [47,48]. Economic disparities are now closely linked to a shift in the food system, with increased availability of generally inexpensive, energy-dense and ultraprocessed food, and have been recognized as one of the main drivers of the current obesity epidemic [11,49,50,51]. Together, it is inevitable to make the association between obesity and an inflammatory state during childhood.

Human adiposity is influenced by complex interactions between genetic, developmental, behavioral and environmental influences [15], and the current scientific literature provides a perspective of previously unsuspected factors contributing to the development of obesity. For instance, the use of broad-spectrum antibiotics in early life has been associated with childhood obesity [52]. Moreover, viral infections have been linked to obesity, particularly by the human adenovirus-36 [53,54]. Despite tangible differences between viral and bacterial infections, they share some major molecular signaling cascades, such as an increment in expression and activation of TLR-4 and SAA pathways [55,56], conceivably indicating the link between acute inflammation and the development of obesity.

## 5. Conclusions

In conclusion, our data describe that conditions leading to acute inflammation may result in long-term negative effects, priming the adipose tissue to generate proinflammatory mediators and to promote cellular proliferation, ultimately predisposing to hypertrophy and insulin resistance. If confirmed in humans, infections may contribute to obesity and type 2 diabetes epidemics when associated with a western diet.

## Figures and Tables

**Figure 1 pharmaceutics-14-00623-f001:**
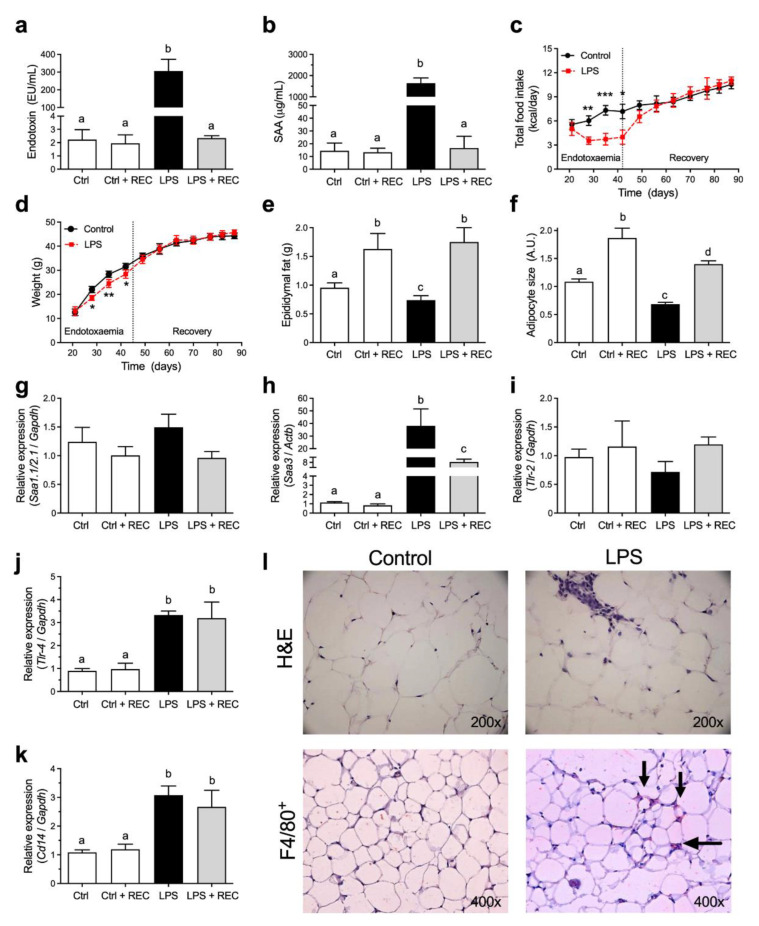
Acute endotoxemia affects adipose tissue but does not lead to weight gain in mice under chow diet. Swiss Webster mice were submitted to intraperitoneal administration of 8 consecutive doses of 10 mg/kg LPS, every 3 days. (**a**) Endotoxin and (**b**) SAA concentration in serum. (**c**) Daily caloric intake. (**d**) Weight gain curve of control and LPS mice. Vertical dashed line in c and d indicate end of acute endotoxemia period. (**e**) Epididymal fat pad weight. (**f**) Epididymal adipocyte size. (**g**–**k**) Quantitative real-time PCR was performed to assess mRNA expression of (**g**) *Saa1.1/2.1*, (h) *Saa3*, (**i**) *Tlr-2*, (**j**) *Tlr-4* and (**k**) *Cd14* in epididymal adipose tissue. (**l**) Histological sections of epididymal fat pads after LPS challenges showing adipocyte morphology on hematoxylin and eosin staining and macrophage infiltration (F4/80^+^). Statistical analysis; for all variables with the same letter (a, b or c), the difference between the means is not statistically significant. Where two variables have different letters (a, b or c), they are significantly different (*p* < 0.05). Data are means ± SEM from 6 mice per group (* *p* < 0.05, ** *p* < 0.01, *** *p* < 0.001, between groups, as indicated).

**Figure 2 pharmaceutics-14-00623-f002:**
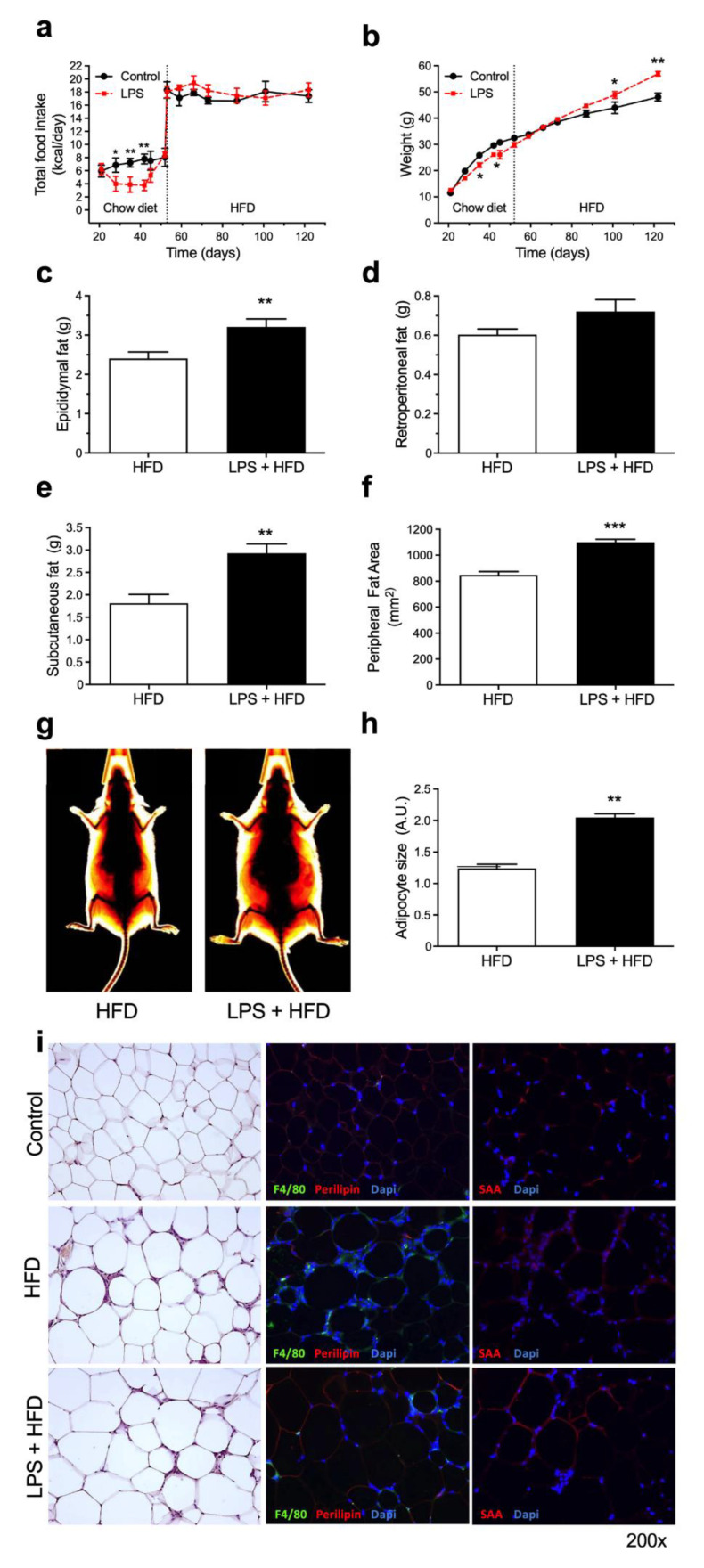
A previous history of acute endotoxemia potentiates weight gain induced by HFD. Swiss Webster mice were submitted to intraperitoneal administration of 8 consecutive doses of 10 mg/kg LPS, every 3 days, followed by 10 weeks in HFD. (**a**) Daily caloric intake, including diet switch after acute endotoxemia period (vertical dashed line). (**b**) Weight gain curve of HFD and LPS + HFD groups. (**c**) Epididymal, (**d**) retroperitoneal and (**e**) subcutaneous fat pad weight after HFD period. (**f**) Subcutaneous fat area quantification in HFD and LPS + HFD mice after HFD period. (**g**) Representative fat area quantification. (**h**) Adipocyte size after HFD period. (**i**) Histological sections of epididymal fat pads after HFD periods showing adipocyte morphology on hematoxylin and eosin staining, macrophage infiltration (F4/80^+^) and SAA production. Data are means ± SEM from 8 mice per group (* *p* < 0.05, ** *p* < 0.01, *** *p* < 0.001, between groups, as indicated).

**Figure 3 pharmaceutics-14-00623-f003:**
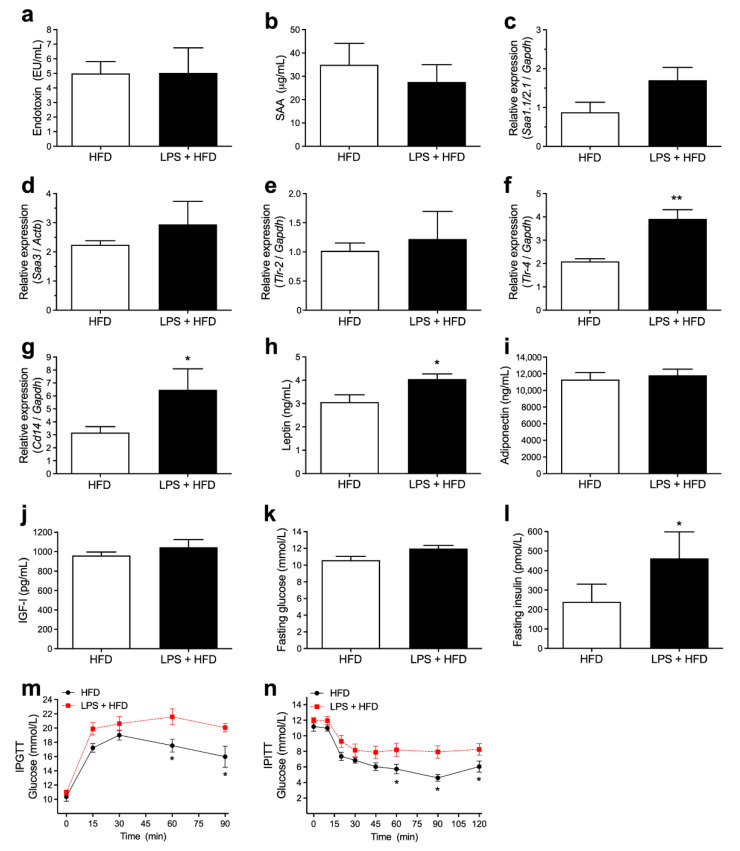
A previous history of acute endotoxemia potentiates glucose tolerance and insulin resistance induced by HFD. After the HFD period, mice previously submitted to multiple acute endotoxemia were evaluated regarding its metabolic parameters. Determination of (**a**) endotoxin and (**b**) SAA in serum. (**c**–**g**) Quantitative real-time PCR for mRNA expression of (**c**) *Saa1.1/2.1* (**d**) *Saa3*, (**e**) *Tlr-2*, (**f**) *Tlr-4* and (**g**) *Cd14* in adipose tissue. (**h**–**l**) At last, serum measurement of (**h**) leptin, (**i**) adiponectin, (**j**) IGF-I, (**k**) fasting glucose and (**l**) insulin. (**m**) Intraperitoneal glucose tolerance test (IPGTT) and (**n**) intraperitoneal insulin tolerance test (IPITT). Data are means ± SEM from 8 mice per group (* *p* < 0.05, ** *p* < 0.01, between groups, as indicated).

**Figure 4 pharmaceutics-14-00623-f004:**
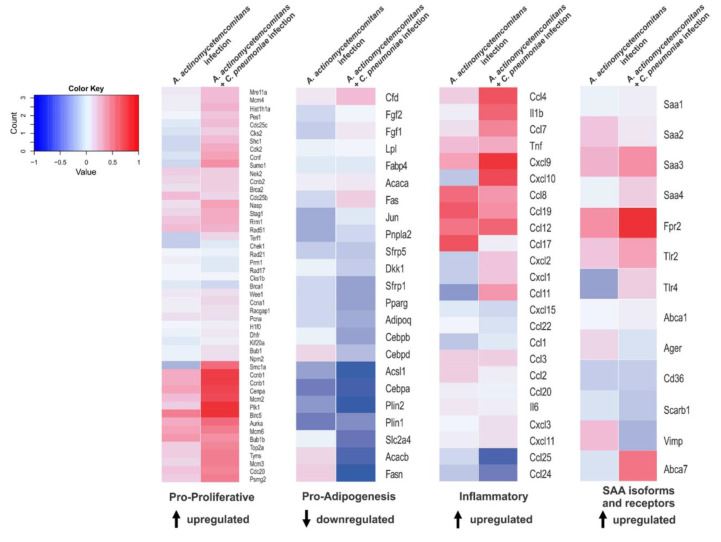
Recurrent infection modulates proliferative, adipogenic, inflammatory and SAA-related genes in adipose tissue. Gene set enrichment analysis (GSEA) revealed that changes in expression of proliferative, adipogenic, inflammatory and SAA-related gene sets in mouse adipose tissue were significantly associated (nominal value of *p* < 0.05) with infection with *A. actinomycetemcomitans* or coinfection with *A. actinomycetemcomitans* and *C. pneumonia* (see methods for details). Heat maps show the mean log_2_ fold-change of all genes of each gene set on each condition compared to uninfected mice.

## Data Availability

The data presented in this study are available in the article.

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
