# Peer review of "Acute Inflammation Is a Predisposing Factor for Weight Gain and Insulin Resistance"

_pharmaceutics, 2022, doi:10.3390/pharmaceutics14030623_

Round 1

Reviewer 1 Report

This study aims to investigate serial intense endotoxemia, characterized by an acute phase response and weight loss, which could be an aggravating or predisposing factor to diet-induced obesity (DIO) and associated metabolic impairments. In general, the experimental work presented is fine and the results are clearly presented. After a minor revision, this manuscript maybe considers for publication.

Introduction:

  1. Endotoxemia or related regulatory mechanisms (factors) of acute inflammation Please explain more clearly in the introduction, otherwise, it is not clear why those indicators are measured
  2. Please state the purpose of this research more clearly in the introduction.

Materials and Methods:

In this study, there are two animal models. LPS+REC is -fat diet f fed with chow diet for 6 weeks after LPS injection; LPS+HFD is fed with high or 10 weeks after LPS injection. Why are the feeding weeks different?

Discussion: Individual indicators are describing, but lack of integration discussion. The ending conclusion is not well developed. The significance of the study could have been expanded.

Reference: Please check the format of the reference. Please abide by the Pharmaceutics Journal format.

Reviewer 2 Report

This study was designed to investigate if recurrent endotoxemia leading to inflammation in mice may result in post-infection insulin resistance and weight gain after a high-fat diet. I have some comments:

  1. In their discussion, the authors extrapolate their findings to children in low-income countries having a high prevalence of infectious diseases. However, these infections are mostly viral not bacterial. To support their conclusions, the authors cite a publication (nr 37) suggesting that adenovirus may be a cause of obesity; however, this paper is a “Medical Hypothesis” with no experimental proof.
  2. A high-fat diet would lead to weight gain anyway, regardless of infections. Indeed, weight gain with high-fat diet following lipopolysaccharide injections was minimal compared to high-fat diet alone (Figure 1c). Therefore, what is the clinical significance of these findings? In this context, I find the statement of the authors that “acute inflammation should be recognized as an aggravating factor for weight gain and obesity comorbidities” (lines 58-60) rather sound.
  3. It is not clear to this reviewer how excess body weight in adults may be associated with an inflammatory state following recurrent infections in childhood (discussion, lines 343-344). I would agree that recurrent infections and inflammation in childhood/adolescence may be associated with aggravation of insulin resistance, thus accelerating the development of type 2 diabetes in those having 1st degree relatives with this type of diabetes. However, how would this condition lead to obesity, particularly in adulthood?
  4. I see the authors measured leptin, a key hormone in the crosstalk between brain and adipose tissue. What is the explanation of its increased serum levels in the group LPS+HFD vs HFD alone? Furthermore, since IGF-1 was also measured, what is the explanation of non-significant differences in Figure 3j?
  5. Please, correct the syntax and grammatical errors in the manuscript, there are several mistakes.
  6. Please, add a paragraph in the beginning or the end of the manuscript explaining the abbreviations.

Round 2

Reviewer 2 Report

  1. The authors have significantly improved their paper according to this reviewer's suggestions. This is a well-presented manuscript providing experimental proof for this interesting hypothesis.
  2. I am aware that leptin is secreted from the adipose tissue. However, my point was whether this increase has any pathophysiological significance or not. If the authors believe that increased blood levels of leptin were simply a marker of the increased adiposity, this should be stated in the discussion.

Author Response

Thank you very much once again for the opportunity to submit a revised manuscript for your appreciation. We have carefully considered the request made by Reviewer 2 and, as suggested, included a paragraph to highlight the significance of increased leptin serum levels in LPS-treated mice under HFD; the sentence can be found below and changes to the manuscript are highlighted in yellow.

[Line 322-326] Increased serum leptin levels, as a proxy of the degree of adiposity, was also observed in the LPS-HFD mice. Leptin regulates adipose tissue mass through central hypothalamus mediated effects on food intake, and increased serum leptin levels in obesity is often associated with a failure of the feedback loop and central leptin resistance [15].